# Factors Predicting COVID-19 Vaccine Effectiveness and Longevity of Humoral Immune Responses

**DOI:** 10.3390/vaccines12111284

**Published:** 2024-11-15

**Authors:** Engin Berber, Ted M. Ross

**Affiliations:** 1Infection Biology, Lerner Research Institute, Cleveland Clinic, Cleveland, OH 44195, USA; berbere3@ccf.org; 2Florida Research and Innovation Center, Cleveland Clinic, Florida, FL 34986, USA

**Keywords:** COVID-19, SARS-CoV-2, vaccines, humoral immune response, vaccine effectiveness, longevity

## Abstract

The COVID-19 pandemic, caused by SARS-CoV-2, prompted global efforts to develop vaccines to control the disease. Various vaccines, including mRNA (BNT162b2, mRNA-1273), adenoviral vector (ChAdOx1, Ad26.COV2.S), and inactivated virus platforms (BBIBP-CorV, CoronaVac), elicit high-titer, protective antibodies against the virus, but long-term antibody durability and effectiveness vary. The objective of this study is to elucidate the factors that influence vaccine effectiveness (VE) and the longevity of humoral immune responses to COVID-19 vaccines through a review of the relevant literature, including clinical and real-world studies. Here, we discuss the humoral immune response to different COVID-19 vaccines and identify factors influencing VE and antibody longevity. Despite initial robust immune responses, vaccine-induced immunity wanes over time, particularly with the emergence of variants, such as Delta and Omicron, that exhibit immune escape mechanisms. Additionally, the durability of the humoral immune responses elicited by different vaccine platforms, along with the identification of essential determinants of long-term protection—like pre-existing immunity, booster doses, hybrid immunity, and demographic factors—are critical for protecting against severe COVID-19. Booster vaccinations substantially restore neutralizing antibody levels, especially against immune-evasive variants, while individuals with hybrid immunity have a more durable and potent immune response. Importantly, comorbidities such as diabetes, cardiovascular disease, chronic kidney disease, and cancer significantly reduce the magnitude and longevity of vaccine-induced protection. Immunocompromised individuals, particularly those undergoing chemotherapy and those with hematologic malignancies, have diminished humoral responses and benefit disproportionately from booster vaccinations. Age and sex also influence immune responses, with older adults experiencing accelerated antibody decline and females generally exhibiting stronger humoral responses compared to males. Understanding the variables affecting immune protection is crucial to improving vaccine strategies and predicting VE and protection against COVID-19.

## 1. Introduction

Severe acute respiratory syndrome coronavirus 2 (SARS-CoV-2) virus, the causative agent of COVID-19, initiated a global pandemic in late 2019 [1]. The disease is characterized by a spectrum of symptoms, from mild respiratory illness to severe pneumonia and even death [2]. The history of SARS-CoV-2 is marked by a series of major variants that have shaped the COVID-19 pandemic [1,3]. This betacoronavirus was first identified in Wuhan, China, in December 2019, with the initial lineage often referred to as the ancestral strain or Wuhan strain [1,3,4,5]. As the virus spread globally, it acquired mutations, leading to distinct variants [6]. The first Variant of Concern (VOC), Alpha (B.1.1.7), was widely reported in the UK in late 2020 and was followed by Beta (B.1.351) in South Africa and Gamma (P.1) in Brazil [3,4,6,7]. These variants shared mutations, such as N501Y and K417N, that increased affinity for the human ACE2 receptor, enhancing transmissibility [8]. Delta (B.1.617.2), emerging from India, dominated by mid-2021 with increased transmissibility and severity. In late 2021, Omicron (B.1.1.529), first detected in South Africa, possessed numerous spike protein mutations, facilitating immune evasion [3,6,7]. Omicron has since diversified into several sub-lineages, including BA.1, BA.2, BA.4, BA.5, and XBB, and continues to evolve and spread globally [6], as depicted in Figure 1. SARS-CoV-2’s evolution is ongoing, and new variants may emerge.

The viral spike (S) glycoprotein plays an important role in virus entry into host cells [9]. This protein consists of two subunits: the S1 subunit, which includes the receptor-binding domain (RBD) that attaches to the ACE2 receptor on human cells, and the S2 subunit, which is involved in viral fusion with the host cell membrane [10]. The SARS-CoV-2 virus initiates infection when spike binds to the ACE2 receptor on the host cell surface. Following proteolytic activation by host cell enzymes, such as TMPRSS2, the spike undergoes a conformational change, thereby facilitating the fusion of the viral and cellular membranes. This fusion event, the beginning of the infection process, allows the viral genome to enter the host cell [11].

Humoral immunity plays a crucial role in vaccine efficacy and protection [12]. Immunoglobulin G (IgG) plays a vital role in neutralizing the virus, while antigen-imprinted antibody-secreting B cells provide long-term protection [13]. IgG antibodies neutralize the virus by binding to the RBD located inside the S1 spike subunit, thus preventing it from interacting with ACE2 receptors and blocking cell entry. People with higher levels of IgG antibodies are less likely to develop severe COVID-19 or require hospitalization. There is a strong correlation between IgG levels and protection against COVID-19, with neutralizing antibody levels strongly predicting protection [14,15,16]. The neutralization level is closely linked to the concentration of IgG antibodies in the blood and serves as a reliable indicator of protection against symptomatic and severe COVID-19 [13]. However, the correlation between the level of antibodies and protection can vary across different variants [17].

The level of circulating antibodies in the sera is important, as it can reduce viral load via neutralization and potentially prevent reinfection in vaccinated individuals. However, antibody levels can decline significantly over time, making the host vulnerable to reinfection [18]. If a good recall immune response is established in the host, upon exposure to the virus, antigen-specific memory B cells can quickly proliferate and secrete antibodies to combat the infection [19]. Although a gradual waning of serum antibody levels is anticipated in the normal course of immune response, various factors may contribute to a more rapid decline after vaccination. Even with a high antibody level, factors like emerging new variants due to mutational changes in the RBD region of the virus can compromise the vaccine effectiveness (VE) [12,20]. These variants, especially Omicron, have mutations in the RBD of the spike protein that have directly impacted the effectiveness of vaccines developed for the ancestral strain. This ongoing evolution of the virus necessitates the reformulation of vaccines to maintain protection, much like annual updates to influenza vaccines.

To combat COVID-19, the development of effective vaccines and therapeutic interventions was a top priority [21,22]. In response to the global pandemic, the World Health Organization (WHO) authorized several vaccines for emergency use. These included mRNA vaccines, such as Pfizer-BioNTech’s Comirnaty (BNT162b2) and Moderna’s Spikevax (mRNA-1273); a viral vector vaccine from Johnson & Johnson’s Janssen; protein subunit vaccines such as Novavax’s Nuvaxovid and Covovax; and inactivated vaccines like Sinovac’s CoronaVac and Sinopharm’s BBIBP-CorV [23]. Other vaccines, such as the adenoviral vector-based vaccine by AstraZeneca-Oxford (ChAdOx1 nCoV-19), Gamaleya’s Sputnik V, and Bharat Biotech’s inactivated Covaxin (BBV152), are not included in the WHO’s Emergency Use Listing (EUL). However, they have received emergency use authorization from numerous countries around the world [24].

The effectiveness of COVID-19 vaccines has been rigorously evaluated in both clinical trials and real-world settings. While mRNA vaccines, such as those produced by Pfizer-BioNTech and Moderna, were highly effective in preventing severe disease, hospitalization, and death in clinical trials during the early phase of the pandemic [25,26], viral vector vaccines produced by AstraZeneca and Johnson & Johnson also provide significant protection [27,28]. Although the exact rates can vary depending on factors such as vaccine type, variant circulation, and population demographics or comorbidities, these vaccines provide significant protection against virally induced disease [20,29,30]. Over time, the effectiveness of the original vaccines in protecting again infection declines, but booster vaccinations significantly increase protection against infection and severe illness [31,32,33,34,35]. The emergence of new variants demonstrates the need for ongoing vaccine development for newly emerging viral variants. Omicron has acquired more than 30 animo acid differences in the critical RBD region compared to the original Wuhan strain. These new variants are able to escape vaccine-induced immune responses, reducing the effectiveness of vaccines [31]. Therefore, prioritizing formulations that directly target circulating SARS-CoV-2 viral variants while continuously monitoring vaccine effectiveness is needed and essential for maintaining long-term protection of the population against SARS-CoV-2 [36,37].

In this review, the various factors that influence the effectiveness and longevity of the humoral immune response to COVID-19 vaccines will be discussed. These factors include the type of vaccine platform, the presence of pre-existing immunity, the impact of booster doses, and the phenomenon of hybrid immunity resulting from both vaccination and natural infection. In addition, factors such as age, sex, and race, as well as comorbidities, like chronic diseases or immunocompromised states, can affect VE and the durability of antibody responses.

## 2. Vaccine Types and Humoral Response

The effectiveness, peak, and decline time points for each vaccine are listed in Table 1. Inactivated vaccines use whole, killed SARS-CoV-2 virus, which elicits immune responses against all viral proteins. Sinopharm (BBIBP-CorV), Sinovac (CoronaVac), and Baharat (Covaxin) are inactivated with β-propiolactone and adsorbed to alum adjuvant [38,39,40]. For the vaccine effectiveness (VE) of CoronaVac (SinoVac) and BBIBP-CorV (Sinopharm), a ~47% to ~84% reduced infection risk against COVID-19 infection was reported [41,42,43]. People vaccinated with CoronaVac vaccine induced neutralizing antibodies in 100% of vaccinated individuals, reaching peak levels approximately two weeks after the second dose, with levels dropping by 160 days. Following the two-dose regimen, the half-life of anti-RBD IgG titers was observed to be ~66 days [44,45]. Importantly, a third booster dose resulted in neutralizing antibody titers that increased to 100% after two weeks, comparable to the peak levels seen after the initial two doses [44].

Viral vector vaccines like AstraZeneca (ChAdOx1), Janssen by Johnson & Johnson (Ad26.COV2.S), and Sputnik V (Ad26/Ad5) utilize a modified non-replicating adenovirus vector virus to deliver the genetic code for the full-length SARS-CoV-2 spike protein. An increase in thrombosis (blood clot) with thrombocytopenia (low platelet count) syndrome (TTS) incidents reported by individuals after receiving an adenoviral vector-based vaccine raises concerns about adenoviral vector vaccines. The estimate of TTS incidence ranges from ~3 to 16 per million cases for the AstraZeneca vaccine (Vaxzevria) and ~2 to 4 cases per million cases for the J&J (Janssen) vaccine [46,47,48]. The FDA revoked the emergency use authorization of the Janssen vaccine by 2023 in the U.S. due to expired vaccine lots and lack of demand, and J&J decided not to update the vaccine composition [49].

The VE of the AstraZeneca vaccine used in a variety of countries ranges from 70% to 91% against symptomatic infections of COVID-19 [27]. The seroconversion rate among individuals vaccinated with AstraZeneca measures ~99% against the RBD and ~98% against the entire spike protein [50,51]. Anti-RBD IgG levels peak one month (21 to 28 days) after the second vaccination dose and rapidly decline over the next six months after vaccination [50,52,53,54,55]. Although these vaccines generate a significant IgG response, the potential for adverse effects may be a greater concern compared to other inactivated or mRNA vaccines. This heightened risk could be due to the vector itself triggering the production of anti-platelet factor 4 PF4 antibodies, which in turn activate platelets through the Fcγ-receptor II [56].

mRNA vaccines, such as those developed by Pfizer-BioNTech and Moderna, elicit high and relatively durable IgG responses. These vaccines, formulated with lipid nanoparticles (LNPs), introduce mRNA encoding the full-length viral spike protein, resulting in the production of the protein using host cell machinery and triggering a strong immune response. The VE of the BNT162b2 (Pfizer-BioNTech) COVID-19 vaccine is 89%-95% in preventing COVID-19 in clinical trials and real-world results [20,25,57,58,59,60]. The VE of mRNA-1273 (Moderna) COVID-19 vaccine was estimated to be 92–96% in a clinical trial and 93% in real-world pooled studies [57,59,60,61,62]. All seronegative participants had detectable anti-spike IgG after the second dose of the BNT162b2 vaccine, and the maximal antibody titer response was reached between 28 and 42 days following the complete series of vaccinations. A decrease was observed starting from 56 days, became more significant 3 months after vaccination, and lowered considerably at 6 months compared to 1 month after vaccination [63,64,65]. Similar to the BNT162b2 vaccine, mRNA-1273 elicits a higher antibody level, reaching a peak 2 weeks after the second dose and showing a moderate decline around 209 days following vaccination [66,67,68].

**Table 1 vaccines-12-01284-t001:** Vaccine effectiveness (VE) against SARS-CoV-2 variants; IgG peak and decline times across different COVID-19 vaccine types.

**Vaccine Name**	**Vaccine Type**	**VE Against Variants ***			**Peak Time of IgG Levels**	**Time of Antibody Decline**
		**Ancestral (Wuhan) or Alpha (B.1.1.7)**	**Delta (B.1.617.2)**	**Omicron (BA.1)**		
Comirnaty (BNT162b2)-Pfizer-BioNTech	mRNA	95% (89.9–97.3) [25] 93.7% (91.6–95.3) [20] 95.3% (94.9–95.7) [58] ^¥^ 93% (85.0–100.0) [57] 91.2% (87.9–94.5) [59] 89% (87.0–90.0) [60]	88.0% (85.3–90.1) [20] 93% (85–97) [35] 92% (90–94) [60]	65.5% (63.9–67.0) [31] 70% (62–76) [69]	14–42 days after second dose [63] 2 weeks after second dose [66] 1 month [65,70,71]	3 months [63,70] Within 6 months after second dose [64,65,71]
Spikevax (mRNA-1273)-Moderna		94.1% (89.3–96.8) [61] 98.1% (96.0–100.0) [59] 92% (88.0–95.0) [60] 98.4% (96.9–99.1) [62] ^¥^ 93% (89.0–97.0) [57]	95% (91–97) [60] 86.7% (84.3–88.7) [62] 92.2% (91.8–92.6) [72]	75.1% (70.8–78.7) [31] 44.0% (35.1–51.6) [73] 40% (38.6–41.3) [72]	2 weeks after second dose [66] 3 months after first dose [67]	By 6 months post 2nd vaccination [66] 209 days after first vaccination [68] 3 to 6 months after first dose [67]
Vaxzevria (AZD1222/Covishield)-AstraZeneca-Oxford (ChAdOx1)	Adenoviral vector	70.4 (54.8–80.6) [27] 74.5% (68.4–79.4) [20] 91% (62.0–98.0) [60] ^¥^ 67% (54.0–80.0) [57]	67.0% (61.3–71.8) [20] 87% (69–95) [60] 82.8% (74.5–88.4) [31]	48.9% (39.257.1) [31]	21 days after second vaccination [53] 28 days after second dose [54] 21 to 28 days after second dose [55]	44% decline by 6 months compared to peak level [55]
Nuvaxovid or Covovax (NVX-CoV2373)-Novavax	Protein Subunit	89.6% (82.4–93.8) [74] 82.7% (73.3–88.8) [75] 89.7% (80.2–94.6) [76] 79.5% (46.8–92.1) [77] ^†^ 89% (75.0–95.0) [78] 90.4% (82.9–94.6) [79]	82.0% (32.4–95.2) [77] 88% (71–95) [78]	12.1 (−78.6–56.8) [80]	2 weeks after second dose [81]	Decline after ~4–6 months [81]
CoronaVac (PiCoVacc)-Sinovac and Covilo (BBIBP-CorV)-Sinopharm	Inactivated	65.9% (65.2–66.6) [82] 83·5% (65.4–92.1) [83] 51% (22.0–63.0) [41] 67% (65.0–69.0) [41] 65.3% (20.0–85) [41] 46.8% (38.7–53.8) [84] 65.7% (63.0–68.5) [59]	59% (47.5–69.6) [85] 30.8 (17.9–41.6) [86] 52% (39.0–63.0) [87] 59(16.0–81.6) [88]	67% (52.0–78.0) [89] 59% (13.0–80.0) [90] 32% (−70.0–73.0) [87]	28 days [91,92]	160 days [45]

* VE % (95% confidence interval); ^†^ pre-Delta strains; ^¥^ meta-analysis.

Protein subunit vaccine NVX-CoV2373, developed by Novavax, presents the immune system with a recombinant nanoparticle containing the full-length spike glycoprotein formulated with Matrix-M adjuvant. In observer-blinded clinical trials, the effectiveness of the vaccine was measured between ~80 and 90% against symptomatic COVID-19 disease [74,75,76,77,78]. A seroconversion rate of ~98% (5 µg) or 99.6% (25 µg) was observed, with antibody titers peaking at 14 days following the second dose of the vaccination and declining over the next 168 days post vaccination (189 day after first vaccine) [81].

## 3. Pre-Existing Immunity (Pre-Immune vs. Naive)

A crucial factor influencing VE is pre-existing immunity (pre-immunity) to the SARS-CoV-2 virus that develops following exposure through natural infection. This concept has been widely explored in influenza virus vaccinations, where the phenomena of immune imprinting and “original antigenic sin” can impact vaccine-induced response [93,94]. Naive refers to individuals who have not previously been exposed to the specific pathogen before vaccination [95]. Following two COVID-19 vaccinations in a naive host, antibodies peak between two weeks and one month post second dose, and the decline begins within three months post second vaccination [63,64,67]. The amount of both anti-spike and neutralizing antibodies wanes considerably six months after the second vaccination [63,66]. By six months of vaccination, the amount of anti-spike antibody wanes to between ~50% and ~80% of the peak levels [65,67]. Although the amount of antibody wanes over time, they persist at low levels between three and six months compared to peak level after vaccination [68]. In contrast, vaccination in a pre-immune host (hybrid immunity) provides a robust and longer-lasting immunity that prevents 90% of reinfection, even in cases where the initial infection was more than 18 months earlier [96]. This indicates that hybrid immunity elicits higher protective immunity that is maintained for more than a year when boosted by vaccination. Individuals infected with SARS-CoV-2 prior to their first vaccination have a significantly lower risk of breakthrough infections compared to those without prior infection (naive), regardless of whether they received the BNT162b2 (Pfizer-BioNTech) or mRNA-1273 (Moderna) vaccine [97]. A retrospective cohort study in Sweden among recipients of the ChAdOx1 nCoV-19 (Oxford-AstraZeneca), BNT162b2 (Pfizer-BioNTech), or mRNA-1273 (Moderna) vaccines found that pre-immunity conferred a 58% lower risk of SARS-CoV-2 breakthrough infection compared to those with natural immunity (prior infection only) and no vaccination. Pre-immunity was also associated with a 66% lower risk of breakthrough infection compared to natural immunity alone for up to 9 months [98]. Vaccination also produces a stronger immune response: among mRNA vaccine recipients, pre-immune individuals show significantly higher antibody levels along with neutralizing titers (anti-RBD IgG) compared to those who are naive [99]. Findings from these studies suggest that hybrid immunity, which arises from a combination of prior SARS-CoV-2 infection and subsequent vaccination, confers a more robust, durable, and comprehensive immune protection compared to either vaccination or natural infection alone.

## 4. Booster Doses

To address the decline and maintain long-term vaccine efficacy, a booster dose (third dose) is recommended to elevate antibody levels and provide durable protection. The Advisory Committee on Immunization Practices advised and recommended that “any FDA approved or authorized COVID-19 vaccine can be used as the booster dose, at least 6 months since primary vaccination” [100]. A cohort analysis revealed that individuals receiving a booster dose of an mRNA vaccine demonstrated significantly enhanced durability of anti-RBD IgG antibodies compared to those who only completed the two-dose primary vaccination regimen [101]. While initial antibody levels measured shortly after the second and third vaccinations (between 7 and 31 days) were comparable, late responses, which were assessed between 90 and 150 days post vaccination, were markedly higher in the boosted group. This finding underscores the critical role of booster immunizations with a monovalent vaccine in establishing a more sustained and robust antibody response against the ancestral strain [101]. Consistent with these results, neutralizing antibody titers decline below 50% within six months after the second vaccination. However, following a third vaccination with either BNT162b2 or mRNA-1273, titers are elevated and remain above 60% for a longer duration, between 6 and 8 months post booster [102]. A large cohort study compared antibody waning after the second and third vaccine doses. Results showed that compared to just two doses, the third dose slowed the decline of both antibody response and neutralizing titers within the first six months. However, when comparing neutralizing titers against Omicron, Delta, and the ancestral strain, Omicron demonstrated higher resistance. This increased resistance was linked to breakthrough infections in boosted individuals, suggesting that while booster doses offer protection against the ancestral and early strains, their effectiveness against Omicron is limited [103]. Similar findings have been observed in people who received a booster dose of the inactivated CoronaVac (Sinopharm) vaccine. Furthermore, an increase in antigen-specific cellular (CD4+ T) immune responses has also been reported following the booster dose [104]. In Table 2, we list the effectiveness against Omicron of various COVID-19 vaccines formulated with the ancestral monovalent strain, along with their updated formulations and booster strategies. The CDC has recommended either homologous or heterologous booster doses to combat emerging variants [100]. Studies have demonstrated that Pfizer-BioNTech’s and Moderna’s homologous boosters show a VE of approximately 67% against Omicron. However, homologous boosters with AstraZeneca, J&J, and inactivated vaccines (CoronaVac or Sinopharm) show VEs of 38–55.6%, 29.4–54%, and 8.6–44.9%, respectively, against Omicron. In contrast, heterologous boosters with any mRNA vaccine demonstrate VEs of 62.4–70.1%, 79%, and 56.8% against Omicron in people previously vaccinated with AstraZeneca, J&J, and inactivated vaccines (CoronaVac or Sinopharm), respectively [31,105,106,107,108]. This indicates that vaccine effectiveness against Omicron or other variants can be improved through heterologous boosters for individuals who received inactivated vaccines and for those without access to new vaccine formulations, such as AstraZeneca and J&J recipients.

## 5. Hybrid Immunity

Hybrid immunity refers to an immune response developed by a combination of natural infection and vaccination. Hybrid immunity occurs when a person become infected before vaccination or when a person experiences a breakthrough infection after vaccination, blending their immune system responses to both the vaccine and the actual virus [109]. Hybrid immunity combines the benefits of both vaccination and natural infection [110]. Boosting immunity confers longer durable protection against symptomatic and severe infection with SARS-Cov-2, although this booster-acquired immunity lasts for several months [101,102,103,104]. Individuals without prior infection that completed the primary vaccination series, and then received a booster vaccination with the BNT162b2 vaccine (3X vaccinated), had a ~52% chance of protection against severe illness following a subsequent SARS-CoV-2 infection [111]. In contrast, for individuals with a history of infection with non-Omicron variants who completed the primary series of BNT162b2 vaccinations (infected plus 2X vaccinated), the effectiveness of protection against severe disease was 55%. This effectiveness increased to ~77% with the addition of a booster vaccination (within 8 months after the second dose) in pre-infected participants (infected plus 3X vaccinated) against both BA.1 and BA.2 variants [111]. Similar results were observed with the mRNA-1273 vaccine recipients in the same study [111]. Vaccinated individuals who experience breakthrough infections tend to have an enhanced and more durable IgG response, as the infection acts as a natural booster. However, three immunizations alone with no infection history show a similar VE (>50%) to that observed in pre-immune and twice-vaccinated individuals, suggesting that protection against symptomatic infection, whether induced by previous infection or vaccination, is associated with individual natural immune responses. This has been examined in a study comparing primary infection with Omicron BA.1 to pre-Omicron (non-Omicron) SARS-CoV-2 against BA.2 re-infection in a case-control study. Pre-infection with a BA.1 SARS-CoV-2 viral variant without additional vaccination is associated with a 72% reduction against BA.2 re-infection compared to pre-infection with a non-Omicron viral variant (38%) and was even three times higher in vaccinated people with no history of previous infection (46%). Hybrid immunity conferred by pre-infection with a non-Omicron viral variant followed by two vaccinations resulted in a reduction in re-infection with BA.2 viral variants. The number of infected individuals was reduced even further when they were boosted a third time [112]. However, greater protection against reinfection with BA.2 was provided with hybrid immunity via pre-infection with Omicron BA.1, and protection further increased to ~96% after two vaccinations. This shows that hybrid immunity confers substantial immune protection against new variants of Omicron if pre-infection immunity is elicited by a previous Omicron viral infection rather than non-Omicron variant infection [112].

## 6. The Impact of Vaccine Formulation and Updates

The spike protein, especially the RBD region located in S1, is an essential target for neutralizing antibodies [113]. Since the pandemic has started, the emergence of Alpha, Beta, Delta, and, more recently, Omicron variants has increased the concerns around virus escape immunity due to mutational changes in the RBD. With ~45 amino acid changes on the Omicron spike, more than 30 have occurred in the RBD, resulting in higher viral transmission rates. As this region of the SARS-CoV-2 changes, it becomes challenging for antibodies to bind effectively, leading to the reduced effectiveness of antibodies in preventing infections. The continuous emergence of new SARS-CoV-2 variants, particularly those with mutations in the RBD of the spike protein, has led to immune escape and a reduced vaccine efficacy [114,115]. Mutational changes in the S protein of Omicron, N501Y and K417N, lead to an increased binding affinity to the ACE2 receptor, suggesting that the enhanced stabilization of the attachment leads to faster virus infection and results in enhanced transmission compared to other SARS-CoV-2 variants [115,116]. These mutational changes that have occurred in the spike protein of the Omicron variant allow for the escape from neutralizing antibodies that efficiently neutralize Wuhan viruses [114]. As of writing this review, five variants of concern have been identified: the Alpha (B.1.1.7) variant in September 2020; Beta (B.1.351) and Gamma (P.1) in November 2020; Delta (B.1.617.2) in December 2020; and Omicron (B.1.1.529) in November 2021 [3,4,6], with Omicron subvariants including XBB.1.5, JN.1, and JP.1,XBB.1.16 and FLIRT variants including KP.1 and KP.2 (https://gisaid.org/hcov19-variants/; https://covid.cdc.gov/covid-data-tracker/#datatracker-home, accessed on 15 August 2024).

Vaccines developed before the emergence of Omicron are still effective in reducing symptomatic infections, especially the mRNA vaccines (like Pfizer and Moderna). Studies have showed these mRNA vaccines to be 90.9% effective against the Delta variant 2–4 weeks after the second dose. However, the effectiveness drops to 65.5% against the Omicron variant in the same time frame. Against Omicron viruses, the VE wanes significantly to about 8.8% after 25 weeks, indicating a more pronounced immune escape due to Omicron’s higher number of mutational changes [117]. This greater immune escape shows the reduction in vaccine effectiveness, prompting manufacturers to reformulate their vaccines. Vaccine reformulation recommendations have been made by health authorities, such as the CDC and FDA in the U.S., which have advised that vaccine updates should match the currently circulating variants. This approach is similar to the annual update process used for influenza vaccines, where the formulation includes potential strains for the next flu season. Based on these recommendations, both Pfizer and Moderna have reformulated their vaccines several times, with reformulations including a bivalent vaccine targeting the ancestral strain and Omicron variants (BA.4/BA.5) during 2022–2023. Later, these vaccines were updated in 2023–2024 to address the XBB.1.5 variant, and, most recently, a monovalent formulation was introduced to target KP.2 for the 2024–2025 season to prevent symptomatic infections of Omicron subvariants and reduce hospitalization rates [118]. A detailed list of vaccine types and formulations, along with the vaccination schedule, can be found in Appendix A.

For those vaccines not updated, heterologous COVID-19 vaccination booster strategies, such as a regimen combining two doses of an adenovirus vector vaccine with one dose of an mRNA vaccine (heterologous), have been employed. The regimen was >80% effective against SARS-CoV-2 infections caused by the Delta or Omicron variants [119]. This phenomenon was also observed in recipients receiving inactivated CoronaVac vaccines that were boosted with the BNT162b2 mRNA vaccine. The heterologous vaccination strategy resulted in a ~57% VE against symptomatic infection in those primed with inactivated vaccine and boosted with mRNA. In contrast, a homologous booster with inactivated vaccine, which was used for both priming and boosting, had a ~9% VE during the Omicron-predominant period of time [107]. Overall, updating the COVID-19 vaccine formulation is critical to address the effectiveness of the immune response against immune escape variants in timely manner.

## 7. Demographics

The effectiveness and durability of antibody responses following COVID-19 vaccination are influenced by various demographic factors, including age, sex, race, and the presence of comorbidities. These factors modulate both humoral immune responses, thus impacting vaccine efficacy and longevity.

### 7.1. Age

In general, age is a leading factor in diminished immune responses to vaccination or antigen exposure. People over 80 years of age are hyporesponsive in both T cell immune responses and T cell-mediated IgG responses to various antigens and mitogens in delayed-type hypersensitivity tests [120]. Notably, those individuals with reduced immune responses to delayed hypersensitivity tests have significantly higher mortality rates over the next two years compared to those who are not hyporesponsive to hypersensitivity tests [120]. Antibody production in response to antigen exposure declines as individuals age, largely due to changes in both the quality and quantity of T cells [121]. This phenomenon is known as immunosenescence [121]. Several factors may contribute to immunosenescence, including a decline in thymus function or an increase in suppressive and regulatory T cells [121]. However, aged individuals may also have a decline in the production naive B cells from the bone marrow, an accumulation of memory cells due to chronic antigen exposure, or low-grade inflammation known as inflammaging, as well as epigenetic and metabolic changes at the cellular level [122].

Consequently, elderly people often have lower peak antibody levels and a more rapid decline in antibody titers following vaccination compared to younger individuals [123]. In influenza-vaccinated individuals, VE prevented death in 78% of young individuals but only prevented death in 50% of people over 65 years old. Similar findings have been observed in older people vaccinated with COVID-19 vaccines, including mRNA, adenoviral vector, and inactivated vaccines, regardless of the vaccine platform [123]. People from various age groups who were vaccinated with the BNT162b2 vaccine (Pfizer-BioNTech) had lower neutralizing antibody titers compared to those vaccinees between 18 and 45 years of age [64]. The initial immune response, particularly the proliferation and production of antibody-secreting B cells and memory B cells, was slower in older participants (≥60 years old) compared to younger ones (60<), although antibody levels eventually reached similar levels over time. However, a third dose (booster) of vaccine significantly enhanced both antibody and T cell responses against Wuhan as well as Delta and Omicron variants [124]. In aged adults, the VE remains similar across different vaccine platforms. For mRNA vaccines, Pfizer-BioNTech had a VE of ~93% in older adults and an overall VE of 95%, while Moderna had a VE of ~86% in older adults versus ~94% overall. For adenoviral vector vaccines, Johnson & Johnson had a VE of ~66% in older adults compared to an overall VE of ~66%, and AstraZeneca showed a VE of 83.5% in older adults versus 74.0% overall. For protein-based vaccines, Novavax had a VE of 88.9% in older adults versus ~90% overall, and for inactivated virus vaccines, Sinovac showed a VE of ~55% in older adults, with a similar estimated VE in the overall population [123]. Although VE has a higher protection against infection and the severity of disease induced by SARS-CoV-2, antibody levels and neutralizing titers begin to decline more rapidly in older than younger people [64,123]. Older individuals tend to have lower peak antibody levels and experience a faster decline in antibody titers after vaccination. For COVID-19, although vaccines remain effective in protecting older adults (with varying efficacy rates across different vaccine types), the decline in antibody titers is faster compared to younger populations. Booster doses significantly improve immune responses in older individuals, enhancing both antibody and T cell activity, especially against Delta and Omicron viral variants.

### 7.2. Sex

In general, females have elevated type I and type II interferon signaling and antibody production compared to males [125]. There is also higher TLR7 expression in B cells in females, which increases antibody production and protection against Influenza A virus after vaccination. Eliminating TLR7 reduces this sex-based difference in antibody response [126]. This immunological difference is thought to be influenced by hormonal factors, such as estrogen and testosterone, as well as genetic factors, including the presence of two X chromosomes in females [125,127,128]. Females tend to mount higher antibody titers with a more sustained antibody response after COVID-19 vaccination [127,128]. Following vaccination with BNT162b2 (primary series of two doses), females had higher anti-RBD IgG responses compared to males in both young and older age groups [129]. Similarly, when comparing anti-S IgG responses between genders, females consistently had higher responses lasting up to 10 months. Although antibody levels declined in both genders, females maintained a higher response post vaccination [128]. How these immunological-related sex differences result in VE and disease susceptibility issues remains an area of active investigation. Meta-analyses have not found that sex differences impact the vaccine efficacy elicited by COVID-19 vaccines [130]. While some studies suggest that females may have a slightly higher risk of vaccine-related adverse events, these events are typically mild and self-limiting [131].

### 7.3. Race and Ethnicity

Racial and ethnic disparities in COVID-19 infection and disease have been a major concern throughout the pandemic, and these disparities may also extend to vaccine responses. A comprehensive meta-analysis exploring the relationship between ethnicity and clinical outcomes in COVID-19 reported that Black and Asian people had a higher risk of COVID-19 infection compared to White individuals [132]. In addition, people of Asian background had a higher number of admissions to the intensive therapy unit [132]. However, vaccine-induced immune responses elicited in different racial populations are not conclusive. In the U.K., people with South Asian ethnicity had an ~16% increase in antibody levels following vaccination compared to White people. In addition, ~12% of people with Mixed/Multiple/Other ethnicity had an increase in antibody levels compared to White people [133]. Neutralizing antibody titers and cellular immune responses to SARS-CoV-2 following mRNA vaccination were stronger in South Asian healthcare workers compared to their White counterparts [134]. Black participants receiving two ChAdOx1 vaccinations had a lower anti-spike antibody peak titer following the second vaccination [135]. Socioeconomic factors, such as occupational risk factors or genetic variations, may contribute to subtle differences in vaccine responses, although more research is needed to understand the impact of vaccine-induced immune responses across race and genetics. The differences in antibody levels does not translate into significant differences in VE against severe disease or hospitalization in vaccinated individuals when comparing socioeconomic factors. However, there is a clear disparity in vaccine uptake among people with different socioeconomic statuses [136,137,138].

## 8. Comorbidities

A wide range of conditions, such as diabetes, autoimmune diseases, obesity, hypertension, cancer, and kidney diseases, significantly increase the risk of severe illness, complications, and mortality in COVID-19 patients [139]. These comorbidities can also affect the effectiveness and durability of the humoral immune response following COVID-19 vaccination. Individuals with these pre-existing conditions generally have lower peaks in antibody titers and experience faster antibody waning compared to healthy individuals. These underlying health conditions, which reduce antibody levels and a shorten the duration of protection, impair the ability of the immune system to mount a robust response to vaccination. For instance, ~12% of individuals with chronic kidney disease (CKD) were seronegative following vaccination, while ~2% of vaccinated people with CKD were seropositive. Similarly, ~31% of individuals with autoimmune diseases did not develop detectable SARS-CoV-2 antibodies after vaccination, compared to ~5% of seropositive individuals with autoimmune diseases. In contrast, healthy participants had significantly higher seropositive rates following full vaccination [140]. In others, seropositivity among vaccinated individuals was 44%, which was lower than in the general population [141]. Anti-RBD IgG and neutralization titers were lower in people with hypertension compared to healthy controls following inactivated and mRNA COVID-19 vaccinations [142,143]. However, there was no association between hypertension and immune response after vaccination [144]. Obesity is also associated with poor clinical outcomes following infection [145]. The blunted response in obese individuals is likely due to chronic low-grade inflammation and altered immune regulation caused by excess adipose tissue [146]. However, studies have not consistently demonstrated poor immunological responses in individuals with obesity [142,144].

An alteration in immune responses has been reported among people with diabetes [147]. Type 2 diabetics (T2D) have lower antibody responses compared to non-diabetic individuals [147]. Diabetics have lower antibody responses compared to non-diabetic controls following COVID-19 vaccination. Specifically, people with T2D had lower seropositivity rates in both Covishield-ChAdOx1 (91 vs. 99%) and Covaxin (33 vs. 83%) inactivated vaccine recipients after the completion of two vaccinations [148]. In addition, people vaccinated with mRNA vaccines had significantly lower neutralizing antibody titers compared to normoglycemic participants and T2D patients with good glycemic control [149]. Interestingly, mRNA VE was higher (~72%) in people over 65 years of age compared to control non-diabetics (~65%). The VE increased following booster vaccinations compared to controls [150]. Overall, even though diabetes is linked to reduced serological responses after COVID-19 vaccination, the impact varies based on factors such as glycemic control and patient age.

People with solid tumors and hematological malignancies had lower rates of seroconversion following COVID-19 vaccination (85% and 59%). The poor seroconversion outcomes observed in patients with hematological malignancies were also associated with low levels of neutralizing antibodies. Specifically, 56% had detectable antibody titers with neutralizing activity against Wuhan SARS-CoV-2, while only 31% of these vaccinated people had detectable titers with neutralizing antibodies against variants of concern [151]. Among people with solid tumors, ~48% had a neutralizing response against Omicron after two doses that increased to 89% after receiving a booster of the mRNA vaccine, which is comparable to responses observed in healthy individuals [152]. In contrast, the majority of those with hematological malignancies had no detectable neutralizing antibodies against Omicron after two vaccinations; however, ~50% of people had detectable neutralizing antibodies following a third (booster) vaccination [153]. Immunocompromised individuals, such as those undergoing chemotherapy or other immunosuppressive therapies, also had severely diminished humoral immune responses. In these populations, medications used to manage these conditions can further suppress immune function, contributing to the reduced vaccine efficacy. For instance, patients that began chemotherapy within 3 months prior to vaccination had an estimated VE of 57%, while people undergoing endocrine therapy had a VE of 76% following COVID-19 vaccination. In comparison, individuals who had not undergone any systemic therapy within 6 months before vaccination had an 85% VE. These results suggest that immunosuppressed subgroups may still be at risk for COVID-19 in the early period following vaccination, despite receiving the vaccine [154]. Cancer patients receiving cancer treatments like immunotherapy, chemotherapy, or both were able to develop similar levels of antibodies after two doses of the Moderna (mRNA-1273) vaccine at similar rates as people without cancer who were not undergoing cancer treatment. However, additional booster vaccinations may offer stronger protection for those who are vulnerable to lower rates of seroconversion in the presence of such comorbidities [155]. Overall, clinical studies confirm that individuals with comorbidities, particularly those with hematological malignancies, benefit significantly from booster doses.

## 9. Discussion

In this review, various factors that affect the immune response to COVID-19 vaccines, including vaccine types, pre-existing immunity, booster doses, hybrid immunity, demographics, and comorbidities have been explored. Different vaccine platforms induce varying levels and durations of antibody responses. The mRNA vaccines, such as those developed by Pfizer-BioNTech and Moderna, elicit high VE and relatively durable IgG responses [20,25,57,58,59,60,61,62,63,64,65,66,67,68,70,71]. Inactivated vaccines, such as CoronaVac and BBIBP-CorV, have also shown effectiveness, but their VE and antibody longevity may be lower compared to mRNA vaccines [41,45,59,82,83,84]. Viral vector vaccines, such as AstraZeneca, provide significant protection, but concerns about rare adverse events have been raised [20,27,46,47,48,57,60]. Protein subunit vaccines, such as Novavax, present a promising alternative, eliciting high seroconversion rates and durable antibody responses [74,75,76,77,78]. Differences in immune responses may be influenced by the vaccine formulation. mRNA and vector-based vaccines present specific protein molecules, guiding the immune response toward the production of anti-RBD or neutralizing antibodies that counter the viral spike protein responsible for ACE2 receptor attachment and viral entry. In contrast, inactivated vaccines contain whole antigens, eliciting a broader immune response against various viral components. The heightened effectiveness of mRNA vaccines may be due to the extended duration of antigen expression.

It has been estimated that pre-immunity due to SARS-CoV-2 infection varies by location, ranging from 10% to 26.6% by the end of 2020 [95]. Pre-existing immunity, acquired through natural infection, significantly impacts VE via harnessing the natural mechanism of immune response. In particular, individuals with hybrid immunity resulting from both vaccination and natural infection exhibit a more robust and long-lasting immune response compared to those with vaccination-induced immunity alone [96,97,110]. This indicates that prior infection, combined with vaccination, provides a major benefit to individuals.

Demographic disparities may increase hospitalization and mortality following SARS-CoV-2 in males, people with a high BMI index, and older people. Females have stronger immune responses following infection or vaccination compared to males, which may be attributed to hormonal and genetic factors [125,126,127,128,129]. However, differences in immune responses between females and males is not substantially elevated following COVID-19 vaccination. Most likely, age is a significant factor affecting the strength of the immune response, as antibody responses decline with increasing age [123]. This age-related decline is also observed in the duration of the immune response, with older individuals experiencing a faster decline in humoral immunity after COVID-19 vaccination compared to younger individuals [123,124]. The gradual decline in immune response with age, immunosenescence, becomes increasingly prominent throughout life [121]. Immunosenescence results from a complex interplay of factors, including decreased production of hematopoietic stem cells in the bone marrow, reduced numbers of naive B cells, increased frequency of immunosuppressive regulatory cells, and thymic shrinkage, which impairs T cell maturation [120,121,122]. One or a combination of these factor may be responsible for the reduction in vaccine effectiveness following COVID-19 vaccination in elderly individuals.

When the pandemic started, those with comorbidities—especially immunocompromised individuals, cancer patients, and those with diabetes and autoimmune diseases—became a concern after infection with SARS-CoV-2. Booster doses have proven to substantially restore both IgG and neutralizing antibody levels, specifically against immune-evasive variants, establishing a sustained and robust antibody response for future variants [103,104]. Even special groups, such as individuals with comorbidities or elderly individuals, can develop immune responses comparable to those of healthy, younger individuals with the help of booster doses [124].

The continuous evolution of the virus leads to the emergence of new variants with increased fitness and affinity for the ACE2 receptor, resulting in higher transmissibility [114,115,116]. This enhanced fitness is often attributed to mutations in the receptor-binding domain (RBD) of the spike protein, which enables the virus to evade immune responses [114]. Current efforts in virus tracking and the public sharing of isolated virus sequences significantly aid authorities and vaccine manufacturers in reformulating vaccines seasonally or annually [118]. This approach of updating vaccine formulations, already successfully utilized in influenza vaccine development, has shown promising results in controlling future outbreaks of drifting flu strains.

Our review is limited by the available data on humoral immune responses, particularly IgG and neutralization findings. While we primarily discuss humoral immune responses, cellular immunity, which also significantly contributes to VE and long-term protection, might be underrepresented. Another limitation is that VE is reviewed mainly in terms of infection prevention; however, VE against hospitalization might be higher, even when VE against infection is low, as seen with immunity acquired through inactivated vaccines. COVID-19 vaccines are relatively new, and long-term data on antibody durability and effectiveness are still emerging. Our review may rely on studies with follow-up periods of only a few months to a year, limiting our ability to project the duration of protection across various vaccine types. As new variants with immune-evasive properties emerge, findings based on past data, particularly regarding VE, may quickly become outdated. For instance, immune escape mechanisms in recent variants might differ from those observed in earlier strains, which could limit the comprehensiveness of our conclusions regarding overall immunity. Furthermore, different vaccine platforms (e.g., mRNA, adenoviral vector, protein subunit, and inactivated virus) elicit varied immune responses (humoral and cellular), making it difficult to directly predict their effectiveness and antibody durability. This diversity may lead to generalized statements that do not uniformly apply across all vaccine platforms.

## 10. Conclusions

The humoral immune response generated by vaccines has been shown to effectively neutralize SARS-CoV-2 and prevent subsequent infections. Although the natural decline of immune responses over time is expected, VE and the longevity of humoral immunity are significantly influenced by intrinsic factors, such as pre-existing immunity, demographics, and comorbidities, as well as extrinsic factors like vaccine type, booster uptake, and vaccine formulation. While mRNA vaccines have demonstrated the highest durability and effectiveness among vaccine platforms, other vaccines have also provided considerable protection against infection. Maintaining VE in vulnerable populations, such as the elderly and those with comorbidities, remains critical. Overall, vaccine effectiveness is predicted to decline more rapidly in men, the elderly, and individuals with immunosuppression or multiple comorbidities. With the ongoing emergence of variants, booster doses appear to offer more durable protection, though new variants continue to pose a constant challenge. The rise of immune-evasive variants like Omicron shows the necessity for continued vaccine reformulation and booster strategies to sustain immunity. As SARS-CoV-2 evolves, updating vaccine formulations and prioritizing high-risk populations will be essential.

## Figures and Tables

**Figure 1 vaccines-12-01284-f001:**
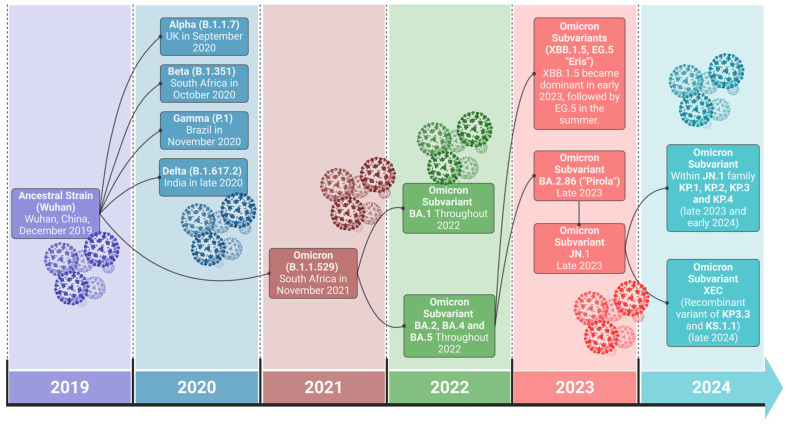
Timeline of major SARS-CoV-2 variants and their evolution from 2019 to 2024. Following the emergence of the original ancestral strain, early variants such as Alpha, Beta, Gamma, and Delta appeared in 2020. The most recent circulating FLIRT variants within the JN.1 family, including KP.1, KP.2, KP.3, KP.4, and the recombinant XEC variant, show the virus is continuing to drive its ongoing evolution. The connecting lines represent the evolutionary emergence of variants from their origins.

**Table 2 vaccines-12-01284-t002:** Vaccine formulation and boosting strategies overview.

Vaccine and Manufacturer	Formulations and Year	Booster Strategy	Effectiveness Against Omicron (with Booster)
Comirnaty (Pfizer-BioNTech; New York, NY, USA-Mainz, Germany)	Monovalent (Ancestral)-2020/2022 Bivalent (Original/Omicron BA.4/BA.5)-2022/2023 Monovalent (XBB.1.5)-2023/2024 Monovalent (KP.2)-2024/2025	Homologous booster doses recommended	67.2% (66.5–67.8) ^η^ [31]
Spikevax (Moderna; Princeton, NJ, USA)	Monovalent (Ancestral)-2020/2022 Bivalent (Original/Omicron BA.1; and Original/BA.4/BA.5)-2022/2023 Monovalent (XBB.1.5)-2023/2024 Monovalent (KP.2)-2024/2025	Homologous booster doses recommended	66.3% (63.7–68.8) ^η^ [31]
ChAdOx1 (AstraZeneca Oxford; Oxford, England)	Monovalent (Ancestral)-2020/2021 No reformulation	Homologous or mRNA booster doses	38.0% (14.2–61.9) ^η^ [105] 55.6% (44.4–64.6) ^η^ [31] 62.4% (61.8–63.0) ^µ^ [31] 70.1% (69.5–70.7) ^π^ [31]
Janssen-Ad26.COV2.S (Johnson & Johnson; Beers, Antwerp, Belgium)	Monovalent (Ancestral)-2021/2021 No reformulation	mRNA booster recommended (heterologous boosting)	29.4% (2.6–56.1) ^η^ [105] 54% (43.0–63.0) ^η^ [106] 79% (74.0–82.0) ^§^ [106]
CoronaVac (Sinovac; Beijing, Haidian District, China; BBIBP-CorV (Sinopharm; Beijing, Haidian District, China)	Monovalent (Ancestral)-2021/2022 No reformulation	Homologous or mRNA booster doses	8.6% (5.6–11.5) ^η^ [107] 36.7% (10.2–63.1) ^η^ [105] 44.9% (19.7–62.2) ^η^ [108] 56.8% (56.3–57.3) ^π^ [107]
Nuvaxovid-NVX-CoV2373 (Novavax; Gaithersburg, MD, USA)	Monovalent (Ancestral)-2021/2022 Monovalent (BA.1)-2022/2023 Monovalent (XBB.1.5)-2023/2024 Monovalent (JN.1)-2024/2025	Homologous or heterologous booster doses	-

^η^ Homologous booster; ^µ^ Moderna mRNA heterologous booster; ^π^ Pfizer-BioNTech mRNA heterologous booster; ^§^ 48% boosted with Pfizer-BioNTech mRNA, and 52% boosted with Moderna.

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
