# Peer review of "Factors Predicting COVID-19 Vaccine Effectiveness and Longevity of Humoral Immune Responses"

_vaccines, 2024, doi:10.3390/vaccines12111284_

Round 1

Reviewer 1 Report

Comments and Suggestions for Authors

General comment

The paper “Factors Predicting COVID-19 Vaccine Effectiveness and Lon-2 gevity of Humoral Immune Responses” is an interesting review about

the factors that influence the effectiveness and longevity of the humoral immune response to COVID-19 vaccines. The article is well written and only needs minor editorial revision.

Specific comments

1)      Abstract. Please. be more explicit about the objective and methods,  “here we discuss….” Is not enough.

2)      Review the end of the Formulations section and be more explicit with the recommendations.

3)      Clarify or correct minor errors: page 9, line 318 “a s of..”; page 11, 426 line “elicited”; page 13, lines 451-452 “For instance,…”).

Author Response

Author's Reply to the Review Report (Reviewer 1)

Comments 1:  Abstract. Please. be more explicit about the objective and methods,  “here we discuss….” Is not enough.

Response 1: Thanks for your attention, we have now added a statement where we highlighted our objective and methods in the abstract.

Comments 2: Review the end of the Formulations section and be more explicit with the recommendations.

Response 2: We have provided a review and recommendation statement to the end of the Formulations section (now changed to Vaccine Formulation Update as requested by reviewer 2nd).

Comments 3: Clarify or correct minor errors: page 9, line 318 “a s of..”; page 11, 426 line “elicited”; page 13, lines 451-452 “For instance,…”).

Response 3: Thanks for your careful reading and attention, now we have corrected all errors pointed in our revised draft.

Reviewer 2 Report

Comments and Suggestions for Authors

The manuscript by Engin Berber and Ted M. Ross entitled "Factors Predicting COVID-19 Vaccine Effectiveness and Longevity of Humoral Immune Responses" requires major changes.

1.      L71-76.- SARS-CoV-2 neutralisation by IgG cannot be generalised, it depends on the variant (Tsuchiya, K. et al. https://doi.org/10.1038/s41598-023-28591-3).

2.      L126.- In section 2. Vaccine types and humoral response, recurring concepts. In addition, many other concepts are repeated throughout the document.

3.      Please list in a table components/ingredients or adjuvant of each vaccine that could be related to the potentiation of the immune response.

4.      The section on vaccine formulation does not refer to formulation.

5.      In discussion, the limitations of the manuscript are not mentioned.

Author Response

Author's Reply to the Review Report (Reviewer 2)

Comments 1:  L71-76.- SARS-CoV-2 neutralisation by IgG cannot be generalised, it depends on the variant (Tsuchiya, K. et al. https://doi.org/10.1038/s41598-023-28591-3).

Response 1: We agree on this comment and appreciate providing a ref. Now we have addressed this in our revised draft.

Comments 2: L126.- In section 2. Vaccine types and humoral response, recurring concepts. In addition, many other concepts are repeated throughout the document.

Response 2: In Section 2, we aim to introduce the types of vaccines and their effectiveness (VE). While this approach may lead to recurring concepts, we wanted to retain the topic on defining each vaccine type and its VE. In the revised draft, we have modified and removed some parts of this section to reduce repetition.

Comments 3:  Please list in a table components/ingredients or adjuvant of each vaccine that could be related to the potentiation of the immune response.

Response 3: Thank you for the improvement. We have supplemented Table S1 in the revised manuscript with a comprehensive list of vaccine types, formulations (including ingredients), and vaccination schedules.

Comments 4: The section on vaccine formulation does not refer to formulation.

Response 4: We have revised the title to “The Impact of Vaccine Formulation and Updates”

Comments 5: In discussion, the limitations of the manuscript are not mentioned.

Response 5: Thanks for pointing this, we have included the limitations of our reviews in the end of the discussion.